# Molecular Mechanisms of Lipid-Based Metabolic Adaptation Strategies in Response to Cold

**DOI:** 10.3390/cells12101353

**Published:** 2023-05-10

**Authors:** Gang Wu, Ralf Baumeister, Thomas Heimbucher

**Affiliations:** 1Bioinformatics and Molecular Genetics, Faculty of Biology, University of Freiburg, 79104 Freiburg, Germany; 2Center for Biochemistry and Molecular Cell Research, Faculty of Medicine, University of Freiburg, 79104 Freiburg, Germany; 3Signalling Research Centres BIOSS and CIBSS, University of Freiburg, 79104 Freiburg, Germany

**Keywords:** lipid metabolism, cold adaptation, membrane fluidity, adiponectin receptor, nuclear hormone receptors, hibernation, mitochondria

## Abstract

Temperature changes and periods of detrimental cold occur frequently for many organisms in their natural habitats. Homeothermic animals have evolved metabolic adaptation strategies to increase mitochondrial-based energy expenditure and heat production, largely relying on fat as a fuel source. Alternatively, certain species are able to repress their metabolism during cold periods and enter a state of decreased physiological activity known as torpor. By contrast, poikilotherms, which are unable to maintain their internal temperature, predominantly increase membrane fluidity to diminish cold-related damage from low-temperature stress. However, alterations of molecular pathways and the regulation of lipid-metabolic reprogramming during cold exposure are poorly understood. Here, we review organismal responses that adjust fat metabolism during detrimental cold stress. Cold-related changes in membranes are detected by membrane-bound sensors, which signal to downstream transcriptional effectors, including nuclear hormone receptors of the PPAR (peroxisome proliferator-activated receptor) subfamily. PPARs control lipid metabolic processes, such as fatty acid desaturation, lipid catabolism and mitochondrial-based thermogenesis. Elucidating the underlying molecular mechanisms of cold adaptation may improve beneficial therapeutic cold treatments and could have important implications for medical applications of hypothermia in humans. This includes treatment strategies for hemorrhagic shock, stroke, obesity and cancer.

## 1. Introduction

Environmental cold causes multiple challenges for organisms. Generally, low temperature slows down the rate of molecular processes and enzyme activities that are essential for survival. Organisms have evolved different adaptation strategies for cold environments. They either increase thermogenesis to keep their core temperature constant (are endothermic) or they are unable to actively regulate their internal temperature and take on the ambient temperature (are ectothermic). Organisms with a variable internal temperature have developed protective physiological adaptative responses to survive in cold conditions. For both cold survival strategies, namely, active temperature regulation through thermogenesis and physiological adaptation due to a variable internal temperature, alterations in lipid metabolic processes, including lipid catabolism and membrane fluidity regulation, are essential.

By increasing their lipid-dependent energy expenditure, homeothermic animals, such as mammals, are able to maintain their core body temperature during cold exposure. Lipids are primarily stored in the adipose tissue in homeotherms and serve as metabolic fuel. To preserve their core body temperature in cold environments, homeothermic animals oxidize lipids in mitochondria predominantly in their brown adipose tissue. During this process, referred to as non-shivering thermogenesis, the chemical energy stored in lipids is utilized to generate heat in mitochondria via uncoupling proteins, which uncouple the electron transport from the respiratory chain [1,2]. When seasonal temperatures are decreasing, several orders of mammals are able to lower their internal temperature (become heterothermic) and hibernate. Hibernating animals have evolved metabolic strategies to preserve energy and decrease their core body temperature to enter an energy-saving torpid state, which results in metabolic repression and a shift from carbohydrates to lipid catabolism [3,4].

Homeothermic organisms predominantly adjust their metabolism to seasonal changes. However, poikilotherms, which have a variable internal temperature according to the ambient temperature, are affected by diurnal temperature fluctuations. Diurnal temperature changes are metabolically challenging, especially for small poikilotherms or microorganisms. They have evolved physiological adaptation processes primarily for their membranes. In poikilotherms, the membrane lipid composition is altered to maintain the optimal membrane fluidity critical for the proper function of membranes in low-temperature conditions [5,6,7]. Such a conservation process of the physiological state of membranes in cold environments is known as homeoviscous adaptation and was first identified in bacteria [8]. In addition, homeoviscous adaptation enables low-temperature survival of poikilothermic species, including nematodes and flies [5,6,9]. It typically leads to an increase of unsaturated fatty acids in membrane phospholipids, which promotes membrane fluidity and counteracts the membrane rigidifying effects of cooling. However, changes in membranes that increase their fluidity are complex and also depend on the fatty acid composition, their chain length and modifications of their head groups [10].

Here we review how changes in membrane properties following cold exposure are detected by membrane fluidity sensors and how such changes are translated into transcriptional outputs altering lipid metabolism. In particular, we focus on the role of the adiponectin receptor in cold sensing and the downstream functioning nuclear hormone receptors of the PPAR (peroxisome proliferator-activated receptor) subfamily, which are master regulators for lipid metabolism. We summarize the literature for regulatory pathways that control lipid metabolic remodeling and are mediated by nuclear hormone receptors, along with their implications on hibernation and hibernation-derived therapies for hemorrhagic shock and stroke. Moreover, the roles of lipid metabolism in mitochondrial-based thermogenesis and mitochondrial dynamics in response to cold are reviewed and their impact on human diseases are discussed.

## 2. Regulation of Membrane Fluidity in Poikilothermic and Cold-Adapted Organisms

A reduction in the environmental temperature has a pronounced effect on the physical properties of membranes, their functions and, ultimately, on the survival of poikilotherms. Membrane lipid bilayers are predominantly fluid at physiological temperatures, which is critical for normal cellular functions [11]. During a temperature decrease, membrane bilayers can change from a disordered fluid to a gel-like non-fluid state [12]. In the non-fluid condition, saturated fatty acyl chains of phospholipids are in a closely packed, ordered arrangement. Consequently, during cold exposure, an excess of saturated fatty acids (SFAs) in phospholipids rigidifies the membrane due to their straight acyl chains, which are stabilized by hydrophobic interactions [13]. Higher-ordered fatty acyl chains are usually in their fully extended conformation, which increases the thickness of the fatty acyl chain area and the distance between polar head groups of the bilayer [14,15,16]. Therefore, a reduced membrane fluidity can result in an elevated membrane thickness under low-temperature conditions. To maintain fluidity and thickness of the bilayer in an optimal range, poikilothermic organisms have developed response mechanisms that can activate lipid desaturases to convert SFAs to unsaturated fatty acids (UFAs). Lipid desaturases introduce double bonds in fatty acids [17], which generate kinks into otherwise straightened acyl hydrocarbon chains of phospholipids. Such double bonds, especially *cis*-double bonds, result in looser packing and increased fluidity of membrane bilayers to maintain their biological functions following temperature downshifts.

Adaptive processes that regulate membrane function were predominantly studied in mesophilic organisms, which prefer to grow at moderate temperatures in a range from 20 °C to 45 °C. However, special adaptation strategies have been evolved by microorganisms thriving in permanently cold ecosystems, the deep sea, and polar or glacial habitats. Such organisms, known as psychrophiles (“cold-loving” organisms), prefer an optimal growth temperature at ~15 °C or below [18] and are often exposed to diurnal temperature changes and repeated freeze and thaw cycles in terrestrial environments. Therefore, they have evolved remarkable strategies to maintain their membrane function under extreme temperature conditions. Physiological adaptations of membranes to cold were comprehensively studied in psychrophilic microorganisms. Psychrophilic bacteria and cyanobacteria increase the proportion of UFAs and short-chain fatty acids (SCFAs) in their membranes [10]. In addition, the head groups of phospholipids and the membrane content of branched-chain fatty acids (BCFAs) are modified to adapt to permanently cold habitats. UFAs are generated by de novo fatty acid (FA) synthesis. Alternatively, double bonds can be introduced into SFAs after their biosynthesis [11], which enables a rapid response to temperature downshifts. Swift desaturase-based membrane modifications are also employed by psychrotolerant bacteria, which have an optional growth temperature of 20 °C to 25 °C but can survive at temperatures below 0 °C [18,19].

Double bonds are usually introduced into fatty acids in a *cis*-configuration by desaturases. UFAs in phospholipids with double bonds in a *cis*-configuration elevate membrane fluidity more efficiently than *trans*-UFAs because the *cis*-configuration results in an immobile 30° kink in the acyl chain [20,21]. The kink causes steric hindrance within fatty acid chains and interferes with the lateral packing of acyl chains in the lipid bilayer. Certain psychrophilic and mesophilic Gram-negative bacteria can regulate an isomerization from the *cis-* to the *trans*-configuration of double bonds in UFAs through a periplasmic enzyme known as *cis–trans* isomerase (Cti) [22,23]. The substrate binding of the isomerase appears to be determined by membrane properties controlling the access of the Cti enzyme to its *cis*-FA substrates located in the inner membrane of Gram-negative bacteria [23]. At low temperatures, the membrane fluidity is reduced, which counteracts an intrusion of Cti into the membrane. However, when the temperature increases and membranes become more fluid, Cti might penetrate the inner membrane bilayer and catalyze the *cis–trans* isomerization of acyl chains. This results in an increase in *trans*-UFAs, which have properties that resemble SFAs and align more closely with each other. Thus, *trans*-UFA generation elevates the viscosity of the membrane to ensure membrane functionality at higher temperatures. The *cis–trans* conversion enables a fast adaptive response (e.g., during diurnal temperature upshifts) and can be employed under growth-inhibiting stress conditions when the fatty acid composition cannot be changed by de novo synthesis [24].

In addition to their acyl chain properties, phospholipids affect the physical state of membranes through their head groups [10,25]. The head groups of diverse phospholipids have different sizes and charges and their acyl chains are differentially modified, which influences the packing and fluidity of the bilayer. In a previous study in yeast using shotgun lipidomics, it was found that *Saccharomyces cerevisiae* alters the proportion of phospholipids in the membrane when exposed to cold [26]. The degree of unsaturation of acyl chains is dependent on the phospholipid class under low-temperature conditions. Such a head-group-specific acyl chain remodeling was recently observed in the Gram-negative bacterium *Methylobacterium extorquens*, which has a relatively simple membrane lipid composition [27]. Following cold exposure of *M. extorquens*, the phospholipids phosphatidylcholine (PC) and phosphatidylethanolamine (PE) display the most pronounced changes in unsaturation. Moreover, the amount of PC lipids in the bacterial membranes increases, whereas PE lipids are reduced during cold conditions. A diminished PE level might counteract the effect of an elevated packing density due to a strong interaction between PE lipids in bacterial membranes [25,27,28]. Conversely, a higher PC content likely improves membrane fluidity at lower temperatures, suggesting that the modulation of phospholipid levels is essential for membrane adaptation in cold.

Psychrophilic bacteria isolated from permanently cold habitats, such as sea ice or arctic glaciers, upregulate the proportion of SCFA and BCFA in their membranes [29,30]. An increase in SCFAs and BCFAs was detected in psychrophilic strains of *Bacillus cereus*, a foodborne pathogen, which can grow in refrigerated food at 4 °C [31]. Short acyl chains of phospholipids do not reach as far into the hydrophobic area of the membrane bilayer as longer acyl chains do. Therefore, shorter chains, especially chains with less than 12 carbons, form weaker hydrophobic interactions with proteins and other lipids, which increases the motion of free acyl chain ends and promotes membrane fluidity in cold environments [32,33]. Contrary to the swift acyl chain remodeling based on desaturation or *cis–trans* isomerization, the incorporation of SCFAs is coupled to bacterial growth because it requires de novo synthesis of fatty acids [32]. De novo synthesis of lipids is also essential for Gram-positive bacteria to upregulate certain BCFAs in response to cold [34]. Methyl branches on BCFAs are predominantly located at the penultimate (*iso*-) or antepenultimate (*anteiso*-) position of fatty acid chains. *Anteiso*-fatty acids in phospholipids have a more pronounced membrane-fluidizing effect than *iso*-fatty acids. The methyl branch in *anteiso*-fatty acids is located further from the end of the fatty acid, which efficiently reduces the packing order of phospholipids’ acyl chains in the membrane bilayer [11]. Psychrotolerant Gram-positive bacteria, such as *Listeria monocytogenes*, increase the proportion of *anteiso*-BCFA and decrease the amount of *iso*-BCFA in the membrane to promote membrane fluidity in response to low growth temperatures [35,36,37]. The regulation of BCFA is species- and temperature-dependent and an upregulation of *iso*-BCFAs is also observed in Gram-positive bacteria when exposed to low-temperature stress [34,38]. Many psychrophilic or psychrotolerant bacteria can replace saturated longer and *iso*-BCFAs with unsaturated shorter and *anteiso*-BCFAs to reduce membrane rigidity as a cold adaptation strategy. Similar responses to cold, namely, an increase in UFAs, SCFAs and BCFAs in membranes, were observed for mesophilic bacteria as well, suggesting that both mesophilic and psychrophilic bacteria appear to share common mechanisms to promote membrane fluidity.

Certain psychrophilic organisms modify their membrane phospholipid pool by increasing the amount of lysophospholipids (LPLs), which are altered phospholipids (PLs) lacking one of their acyl chains. Antarctic psychrophilic yeast strains naturally synthesize increased levels of lysophosphatidylethanolamine (LPE) and lysophosphatidylcholin (LPC) compared with mesophilic yeast *S. cerevisiae* [39]. Membrane LPLs can be generated via hydrolysis of an acyl chain in PLs through the enzymatic activity of phospholipases as part of the de-acylation/re-acylation cycle (Lands’ cycle [40]) or via de novo synthesis of PLs. LPLs were found in membranes of animals in relatively low quantities, e.g., in insects only around 1% of total PLs are LPLs [41,42]; however, their proportion can increase during cold exposure. LPLs have an inverted conical shape and hence disrupt the packing order of PLs’ acyl chains in membranes, which increases membrane fluidity [9]. Elevated LPL levels were detected in *Drosophila* in response to low temperatures and during cold acclimation [41,43]. In addition, LPLs are upregulated during seasonal acclimatization of the bug *Pyrrhocoris apterus* [42]. These studies suggest that LPLs are essential components in membranes for shaping thermal responses. The specific functions of LPLs in cold adaption have only been studied in a small number of organisms so far and are still poorly understood, but might be relevant for cold-related responses of many species in their natural habitats.

## 3. Sensing Membrane Rigidification Is Essential for Membrane Fluidity Maintenance in Cold Adaptation

Poikilothermic organisms can sense a decrease in membrane fluidity and have evolved feedback mechanisms to maintain membrane homeostasis. Membrane-bound sensors that detect membrane rigidification were initially identified in bacteria. Bacteria such as *Bacillus subtilis* and cyanobacteria use an ancient kinase-based sensing system composed of a sensory kinase and a cognate response regulator, which can alter the biophysical properties of membranes by promoting the expression of acyl lipid desaturases [44,45]. In the cyanobacterium *Synechocystis* sp. *PCC 6803*, the histidine kinase Hik33 was identified as a transmembrane cold sensor that signals to a transcriptional response regulator. Hik33 phosphorylates and activates the response regulator and induces the expression of fatty acid desaturases, which act as membrane fluidizers following a reduction in the ambient temperature [44,46]. The discovery of this two-component sensory system in cyanobacteria demonstrated that the membrane can be the primary site of temperature perception. However, additional studies are needed to further investigate how Hik33 senses changes in the thickness of the lipid bilayer or alterations in the physical motion of membrane lipids when the temperature is reduced.

One of the best-investigated system for cold perception and associated transcriptional responses was characterized in *Bacillus subtilis*, which senses changes in bilayer viscosity through the transmembrane cold sensor DesK. DesK is a multi-pass transmembrane protein, which possesses both a histidine kinase and a phosphatase activity and controls a response regulator for the synthesis of UFA [47,48]. Upon decreasing ambient temperature and the rigidification of membranes, a kinase-dominant state of DesK is promoted, resulting in the phosphorylation and activation of the transcriptional response regulator DesR [49,50]. DesR initiates the transcription of a ∆5 acyl lipid desaturase, which, in turn, causes an increase of desaturated acyl chains in the phospholipids of membrane bilayers [49,51]. A higher proportion of UFA in membranes might trigger a feedback mechanism since an increase in fluidity appears to promote a phosphatase-dominate state of DesK, which causes the dephosphorylation and inactivation of DesR and the termination of lipid desaturase transcription [49,50]. The ability of the cold sensor to switch its activity from a kinase to a phosphatase and vice versa is based on conformational changes and provides a rapid mechanism to adjust membrane properties through fine-tuning the expression of an acyl lipid desaturase when the ambient temperature is fluctuating. The temperature-sensing mechanism of DesK was comprehensively studied. Initially, it was proposed that the transmembrane protein DesK detects changes in the thickness of membranes [52,53,54]. However, in addition to detecting alterations in the bilayer thickness, other membrane properties, such as fluidity, rigidity or permeability to water, might be sensed by DesK [55].

Membrane fluidity sensors were also identified in the nematode *C. elegans*. *C. elegans* is a poikilothermic organism that is approximately 1.3 mm in length, and thus, has a small body volume. A drop in environmental temperature rapidly affects its membrane properties. In worms, the membrane bilayer fluidity is monitored by the multi-pass transmembrane protein PAQR-2, which is a progestin and adipoQ receptor-like protein that is homologous to the mammalian adiponectin receptor AdipoR2 [56]. PAQR-2 is required for cold adaptation and can sense membrane rigidification [57]. Similar to bacterial Hik33 and DesK, PAQR-2’s cold-sensing activity promotes the expression of acyl lipid desaturases, such as ∆9 desaturases, in *C. elegans* [57]. As possible transcriptional regulators for ∆9 desaturases, the HNF4 and PPARα homolog NHR-49 (Figure 1a) and the sterol regulatory element-binding protein/SREBP homolog SBP-1 were suggested to function in a genetic pathway with PAQR-2 [56,57]. SBP-1 and NHR-49 interact with the mediator subunit MDT-15, which is a transcriptional coactivator that associates with RNA polymerase II and is required for ∆9 desaturase expression [58,59]. Gain-of-function alleles of *nhr-49* and *mdt-15*, as well as *sbp-1* overexpression, were identified as suppressors of *paqr-2* loss-of-function phenotypes, suggesting that these regulators might be part of a cold adaptation pathway with PAQR-2 as an upstream cold sensor that controls membrane fluidity via fatty acid (FA) desaturation [57]. In addition, Svensk and colleagues found that *paqr-2* loss-of-function phenotypes are suppressed by a reduction of phosphatidylcholine (PC) synthesis, which, in turn, causes the activation of SBP-1 [60]. Thus, low PC synthesis may indirectly activate SBP-1 to promote ∆9 desaturase expression, causing FA desaturation and cold adaptation [57].

Contrary to DesK, whose ability to sense membrane viscosity depends on its structural domains, PAQR-2 requires the coregulator IGLR-2 (immunoglobulin domain and leucine-rich repeat-containing protein 2) for detecting membrane rigidification (Figure 1a) [61]. IGLR-2 is a single-pass transmembrane protein and associates with PAQR-2 when membrane fluidity is reduced under low-temperature conditions. Membrane rigidification and thickness might result in local clusters of PAQR-2 and IGLR-2 in the lipid bilayer [62]. Such clusters of PAQR-2-IGLR-2 transmembrane proteins with an inflexible length appear to be thermodynamically more favorable if the membrane is thicker and rigid. A thicker membrane is only deformed at the protein cluster site by transmembrane proteins with an inflexible length and does not need to be deformed multiple times for each transmembrane protein locally when transmembrane proteins are distributed unclustered over the membrane. IGLR-2 primarily interacts with transmembrane domains of PAQR-2, which might cause conformational changes in PAQR-2′s N-terminal cytoplasmic domain (Figure 1a) [63]. Based on crystal structures of human AdipoR1,2 [64] and a PAQR-2 structural prediction by the AlphaFold protein structure database [65,66], PAQR-2′s transmembrane domains form a barrel structure with an opening facing the cytosolic side. Busayavalasa and colleagues propose a model suggesting that the regulatory N-terminal cytoplasmic domain of PAQR-2 usually blocks access to the opening and PAQR-2′s catalytic site (Figure 1a) [63]. In a thicker membrane with reduced fluidity, the association of IGLR-2 with PAQR-2 is stabilized and the cytoplasmic inhibitory domain of PAQR-2 might be displaced by IGLR-2 to allow access to the catalytic site of PAQR-2. Thus, when IGLR-2 activates PAQR-2, PAQR-2′s hydrolyzing activity can convert more substrate to potential signaling products. In yeast, PAQR-2 receptor-like proteins can hydrolyze ceramides through their ceramidase activity to generate free FAs and sphingoid bases [67]. Sphingoid bases act as second messengers and might activate kinases implicated in signal transduction. In summary, PAQR-2 and its binding partner IGLR-2 are thought to act as fluidity sensors and detect membrane rigidification during temperature downshifts. Activated PAQR-2 might initiate a signal transduction cascade to its putative transcriptional response regulators, such as NHR-49 and MDT-15, which control fatty acid desaturase expression and are critical for balancing membrane homeostasis.

**Figure 1 cells-12-01353-f001:**
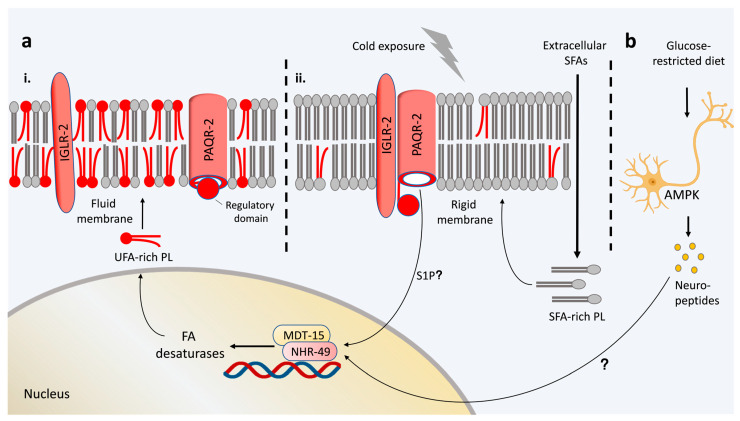
The nuclear hormone receptor NHR-49 functions downstream of the membrane fluidity sensor PAQR-2 and the AMP-activated protein kinase (AMPK) to promote membrane fluidity. (**a**) PAQR-2 and its binding partner IGLR-2, which is a transmembrane protein, act as sensors for low membrane fluidity caused by cold exposure or dietary saturated fatty acids (SFAs). In a proposed model by Busayavalasa and colleagues, PAQR-2′s regulatory domain blocks access to the catalytic site of PAQR-2 in fluidized membrane conditions (i) [63]. Rigidification and thickening of the membrane stabilize the interaction of PAQR-2 with IGLR-2, which, in turn, results in a displacement of the regulatory domain and enables access of a PAQR-2 substrate to PAQR-2′s catalytic site (ii). PAQR-2 has a putative ceramidase activity and might catalyze ceramides to sphingosines. It still needs to be determined whether sphingosine 1-phosphate (S1P), which is generated by the phosphorylation of sphingosine, induces NHR-49. NHR-49 associates with MDT-15 and promotes ∆9 desaturase expression and membrane fluidity. The model for PAQR-2′s membrane-sensing function was adapted from Busayavalasa and colleagues [63,68]. (**b**) A glucose-restricted diet activates a neuronal AMPK variant, resulting in neuropeptide release and activation of NHR-49. NHR-49 function requires PAQR-2 activity in a glucose-restricted dietary regimen to induce membrane fluidity via ∆9 desaturases [69]. Red-colored phospholipids indicate phospholipids with double bonds in their acyl chains. PAQR-2, progestin and adipoQ receptor-like protein; IGLR-2, immunoglobulin domain and leucine-rich repeat-containing protein 2; NHR-49, nuclear hormone receptor-49; MDT-15, subunit of the Mediator complex; PL, phospholipids; UFA, unsaturated fatty acids.

## 4. Mammalian Adiponectin Receptors Signal to Downstream Lipid Regulators

Adiponectin receptors (AdipoRs) are transmembrane receptors and were initially identified as receptors for the adipokine adiponectin in mammals [70]. Adiponectin is expressed and secreted by the adipose tissue [71,72]. It promotes fatty acid catabolism, increases insulin sensitivity in the liver and inhibits hepatic gluconeogenesis [73,74,75]. Adiponectin levels are decreased in mouse models for obesity and type 2 diabetes, whereas administration of adiponectin is considered as a therapeutic treatment strategy for the metabolic syndrome associated with obesity [76,77]. Cold exposure of mice and humans elevates plasma adiponectin concentrations, and adiponectin is essential for subcutaneous adipose browning in mice [78,79,80]. Moreover, adiponectin is involved in thermogenesis and core body temperature regulation in cold environments, although conflicting results were obtained for body temperature control in adiponectin knock-out mouse models [81,82]. Mammalian adiponectin exerts its functions via binding to the seven-transmembrane adiponectin receptors, namely, AdipoR1 and AdipoR2 [70]. AdipoR1 is relatively ubiquitously expressed throughout all tissues and most abundant in skeletal muscle, whereas AdipoR2 is predominantly expressed in the liver [70,77]. Double knock-out of both AdipoR1 and AdipoR2 abrogates adiponectin binding in mice, suggesting that adiponectin receptors are essential for mediating adiponectin’s function in mammalian lipid and glucose metabolism [83].

Adiponectin receptors additionally act as membrane fluidity sensors in mammalian cells. Cell-culture-based knock-down of both adiponectin receptors, namely, AdipoR1 and AdipoR2, causes an accumulation of SFAs in membrane phospholipids and a rigidification of the membrane bilayer [84]. AdipoR2 deficiency has a more pronounced effect on membrane fluidity than the depletion of AdipoR1. This reflects the situation in *C. elegans*, which has two partially redundant AdipoR homologs, namely, PAQR-1 and PAQR-2; however, only PAQR-2 plays a dominant role in cold adaptation and membrane homeostasis. Similar to *C. elegans* PAQR-2, a loss of AdipoR2 results in a reduced expression of fatty acid desaturases, including fatty acid desaturases 1 and 2 (FADS1, FADS2) and stearoyl-CoA desaturases (SCDs) [85]. In addition to regulating FA desaturation, a loss of AdipoR2 affects the transcription of genes controlling cholesterol biosynthesis and acyl chain remodeling of phosphatidylcholines for maintaining membrane homeostasis when cells are challenged with SFAs, such as palmitic acid (Figure 2). Membrane-rigidifying effects of AdipoR2-deficient cells can be reversed by supplementing membrane-fluidizing monounsaturated fatty acids (MUFAs) or polyunsaturated fatty acids (PUFAs), suggesting that the primary function of AdipoR2 is to detect and antagonize membrane rigidity [85,86]. Interestingly, a glucose-rich diet for *C. elegans* promotes the saturation of fatty acids and, therefore, increases membrane rigidity [61,68]. Under such a dietary condition, PAQR-2 is required to readjust membrane fluidity to alleviate the membrane rigidifying and toxic effect of glucose. Conversely, a glucose-restricted diet based on an *E. coli* glucose-depleted mutant improves *C. elegans* healthspan by promoting membrane fluidity [69]. The beneficial effect of a glucose-restricted diet is mediated by a neuronal version of the AMP-activated protein kinase (AMPK) and requires PAQR-2 and its downstream effector NHR-49 (Figure 1b). Neuronal AMPK functions non-cell autonomously via a putative neuropeptide to promote NHR-49 and ∆9 desaturase activities to enhance membrane fluidity. These data illustrate that the adiponectin receptor PAQR-2 and its downstream functioning nuclear hormone receptor NHR-49 are essential to integrate dietary impacts for balancing membrane homeostasis.

Adiponectins, which are the proposed activating ligands of adiponectin receptors, were increased in the serum of mice during a 4 °C cold challenge [87] and in human plasma when healthy male individuals were exposed to cold conditions for 2 h [78]. However, recent cell-culture-based studies indicated that adiponectin is not required for AdipoR-mediated membrane fluidity maintenance [86]. Similarly, an adiponectin homolog has not been found in *C. elegans* yet, and genetic screens for the suppressors of *paqr-2* loss-of-function phenotypes failed to identify ligands for PAQR-2 [86]. Taken together, nematode and mammalian adiponectin receptors are crucial regulators for membrane fluidity, but it appears that they might mediate their sensing function for membrane rigidification without binding to adiponectin.

The signaling pathways downstream of AdipoRs were comprehensively investigated in mouse models. Studies in knock-out mice suggested that AdipoR1 mediates its metabolic effects in mouse liver via AMP-activated protein kinase (AMPK) stimulation, whereas the peroxisome proliferator-activated receptor α (PPARα) functions downstream of AdipoR2 (Figure 2) [83,88]. AdipoR1 activates the AMPK pathway to suppress lipid synthesis, and AdipoR2 promotes PPARα activity to stimulate fatty acid oxidation [83]. The AdipoR2–PPARα signaling axis is potentially conserved from mammals to nematodes, as the *C. elegans* adiponectin receptor PAQR-2 acts in a cold adaptation pathway, along with NHR-49, which has functions similar to mammalian PPARα and HNF4α (hepatocyte nuclear factor 4α). This suggests that the nuclear hormone receptor PPARα might be a crucial regulator for cold adaptation in response to adiponectin receptor-mediated cold sensing. PPARα controls diverse processes in lipid metabolism, including the transport of FAs and their degradation through mitochondrial and peroxisomal fatty acid oxidation [89]. The overexpression of AdipoR2 in mouse liver and muscles transcriptionally upregulates PPARα and its targets, such as an acyl-CoA oxidase, which catalyzes the first step in peroxisomal β-oxidation, and an uncoupling protein (Ucp) [83,90]. Mitochondrial Ucps reside in the inner membrane of mitochondria and can mediate a controlled discharge of a proton gradient linked to the oxidation of metabolic fuels. This energy-dissipating process is essential for heat production during thermogenesis when an organism is exposed to cold or hibernates.

Based on structural data, AdipoR1/2 receptors contain an intrinsic ceramidase activity and are able to hydrolyze ceramides to generate free fatty acids and sphingosine (Figure 2) [91]. Sphingosine can be phosphorylated by sphingosine kinases (Sphks) to produce the signaling molecule sphingosine 1-phosphate [92]. The ceramidase activity of an AdipoR2-like receptor was initially described in yeast [67].

In a recent study, it was demonstrated that the ceramidase activity of mammalian AdipoR2 initiates a sphingosine 1-phosphate-based signaling pathway to activate the nuclear hormone receptor PPARγ and the sterol regulatory element-binding protein-1 (SREBP1) [93] (Figure 2). Membrane rigidification promotes AdipoR2′s ceramidase activity, which, in turn, results in the production of the signaling molecule sphingosine 1-phosphate. Sphingosine 1-phosphate stimulates PPARγ and SREBP1 activity and their transcriptional target, namely, a stearoyl-CoA desaturase, to promote the generation of UFAs and membrane fluidity. These findings, along with previously published *C. elegans* studies, suggest that mammalian AdipoR2 and an AdipoR2-like receptor in nematodes function as membrane fluidity sensors and signal to downstream nuclear hormone receptors, such as PPARγ and PPARα, to control lipid metabolism, including fatty acid desaturation for balancing membrane fluidity.

## 5. Lipid Bilayer Stress in the Endoplasmic Reticulum Induces the Unfolded Protein Response

The endoplasmic reticulum (ER) is a central organelle regulating protein and lipid homeostasis in eukaryotes. Proteotoxic and lipid-related perturbations disturb ER function and homeostasis. Aberrant ER function results in ER stress and activation of an ER stress response pathway known as the ER unfolded protein response (UPR^ER^). The UPR^ER^ is induced not only by unfolded proteins but also via an aberrant ER membrane composition [94]. In metazoans, ER stress is detected by three ER stress sensors: Inositol-requiring enzyme-1 (IRE1), protein kinase RNA-like ER kinase (PERK) and activating transcription factor-6 (ATF6). Activation of these UPR^ER^ transducers can result in translational attenuation or transcriptional activation of factors that control ER protein folding and degradation [95,96,97]. UPR^ER^ stress sensors respond to an accumulation of unfolded proteins; however, IRE1 and PERK additionally sense atypical lipid compositions in the ER membrane, which is referred to as lipid bilayer stress. Both IRE1 and PERK are activated by elevated levels of SFA within the ER lipid bilayer [98]. IRE1 can detect additional membrane perturbations, such as an altered PC-to-PE ratio, depletion of inositol and elevated sterol levels [99,100,101]. To sense lipid bilayer stress, the transmembrane domains of IRE1 and PERK are essential [98]. The luminal domain, facing the ER lumen, is dispensable to sense an aberrant ER membrane lipid composition. The luminal domain is usually required to sense proteotoxic stress caused by an overload of unfolded proteins in the ER lumen. In a previous study, a potential bilayer stress-sensing mechanism for yeast IRE1 was identified [102]. In the proposed model by Halbleib and colleagues, increased lipid order, which is linked to lipid bilayer stress, promotes oligomerization and activation of IRE1. An elevated lipid order or membrane rigidity is associated with the cold exposure of an organism and might be sensed by IRE1. In a recent study, IRE1 was found to be activated in neurons under extremely cold conditions to control lipid composition and cold adaptation in *C. elegans* [103]. Similarly to previous studies, the luminal domain of *C. elegans*’ IRE1 was dispensable; however, the transmembrane and cytosolic domains of IRE1 were essential to confer cold stress resistance. Moreover, signaling mediated by the adiponectin receptor AdipoR2 might affect the functionality of ER membranes as well. AdipoR2 deficiency combined with exposure to SFA results in an upregulation of ER stress response genes [85]. The effect of impaired AdipoR2 activity on ER stress is likely indirect. Membrane perturbations and increased membrane rigidity caused by AdipoR2 deficiency might affect ER membrane properties and induce the UPR^ER^. These findings suggest that AdipoR2-mediated signaling likely influences ER integrity indirectly via an impaired lipid membrane homeostasis during SFA-induced lipid stress or cold exposure.

## 6. PPARα Regulates Lipid and Ketone Metabolism in Heterothermic Hibernating Animals

Hibernating mammals are able to decrease their core body temperature (become heterothermic) and enter an energy-conserving torpid state when ambient temperatures drop and environmental food resources are limited. Smaller mammalian hibernators, including ground squirrels and bats, can drastically reduce their body temperature close to ambient conditions during bouts of extended torpor, whereas large hibernators, such as black bears, only moderately diminish their body temperature down to 30–36 °C [104]. In addition, hibernating mammals can lower their metabolic rate and undergo metabolic reprogramming [105,106]. During bouts of torpor, lipid stores become the primary fuel source due to a metabolic switch toward lipid utilization in the absence of feeding [3]. Fat-storing hibernators, which do not depend on food caches, accumulate a large amount of lipids in their white adipose tissue prior to the onset of hibernation by going through a hyperphagic period, resulting in a massive weight gain at the end of the summer season. A shift in fuel utilization to catabolize triglyceride stores during torpor is likely mediated by nuclear hormone receptors [107]. One of the prime nuclear hormone receptors that control lipid metabolic reprogramming is PPARα [89]. PPARα might promote a shift in fuel utilization from both glucose and fatty acids in regular metabolic conditions toward predominantly fatty acids during hibernation, which resembles metabolic changes similar to fasting [108]. However, it needs to be determined whether PPARα is primarily activated by cold or by the starved condition of hibernating animals. A key regulator for a shift in fuel selection from glucose to fat is pyruvate dehydrogenase kinase 4 (PDK4) (Figure 3). PDK4 is a downstream target of PPARα and is transcriptionally upregulated in hibernating ground squirrels [3,4,109,110]. PDK4 phosphorylates and inhibits pyruvate dehydrogenase activity, and thus, promotes a switch from glucose toward fatty acid oxidation [111]. The metabolic rewiring mediated by PDK4 is accompanied by an upregulation of fatty acid β-oxidation during hibernation. Several genes that encode enzymes for peroxisomal and mitochondrial β-oxidation are transcriptionally upregulated in the torpid state [112,113,114,115,116,117]. An increase in lipid catabolism appears to be largely mediated by PPARα since PPARα transcriptionally controls almost all enzymes implicated in the mitochondrial uptake and oxidative breakdown of fatty acids [89], and thus, might represent a crucial regulator of torpid lipid metabolism (Figure 3). PPARα, as well as PPARγ, which is another member of the PPAR nuclear hormone receptor subfamily, are induced along with their shared coactivator PGC-1α (PPARγ coactivator-1α) during hibernation [116,118]. Nuclear hormone receptors (NHRs) typically associate with coactivators and corepressors. Upon stimulation of NHRs, corepressors are replaced by transcriptional activators, which initiate the expression of NHR target genes. PGC-1α is a cold-inducible transcriptional coactivator and activates PPARα to promote adaptive thermogenesis and fatty acid oxidation [119,120].

Additionally, PPARα induces the expression of a key regulator for torpor, namely, the fibroblast growth factor 21 (FGF21). FGF21 functions as an endocrine hormone, stimulating torpor and reducing the core body temperature in fasted mice [121]. Remarkably, FGF21 has an important impact on metabolism during fasting and torpor since it promotes ketogenesis (Figure 3). Ketogenesis is a metabolic prosses that catalyzes the formation of ketone bodies, which are largely generated by breaking down fatty acids in the liver. Blood and tissue concentrations of ketone bodies are significantly elevated during torpor in various studied species [122,123,124] and serve as an alternative fuel source for prolonged fasted or starved animals. As glucose levels are limited during starvation or torpor, ketone bodies are an essential energy source, especially for the brain, which only poorly utilizes fatty acids as fuel [108,125]. Ketogenesis is also promoted by FGF21′s upstream regulator PPARα. PPARα controls both the formation of ketone bodies and their transport during fasting and hibernation. A rate-limiting mitochondrial enzyme in ketone body synthesis, namely, hydroxymethylglutaryl coenzyme A synthase 2 (Hmgcs2), is upregulated during hibernation [126]. *Hmgcs2* is a target of PPARα and can be induced during fasting or via PPARα agonists, along with additional enzymes for ketogenesis [89]. Ketone bodies are produced in the liver and distributed to various tissues through the blood circulatory system. Monocarboxylate transporters (MCTs) facilitate the transport of ketone bodies and other monocarboxylic acids across biological membranes. The PPARα target monocarboxylate transporter 1 (*Mct-1*) is induced at the blood–brain barrier in torpid animals. This suggests that in addition to PPARα’s role in ketone body production, PPARα is also able to promote the uptake of ketones into tissues by upregulating their transporter MCT-1 (Figure 3) [127,128]. Studies in hibernating ground squirrels demonstrated that the ketone D-β-hydroxybutyrate (BHB) is a preferred energy substrate for the hibernating brain, which is usually highly dependent on glucose [127]. Glucose is a relatively limited metabolic fuel during hibernation. The major source of carbon for gluconeogenesis is amino acids in starvation and torpor conditions [122,129]. Thus, a switch from glucose to fat-derived ketone body utilization protects against an extensive erosion of protein mass during hibernation. Moreover, the metabolic shift from glucose toward ketone catabolism in torpor limits lactic acid production in the brain [106]. Carbons from ketones, such as BHB, enter the tricarboxylic acid (TCA) cycle without producing lactic acid. This might protect against tissue damage from lactic acidosis during periods of diminished blood flow in hibernating animals.

Tissues in hibernating animals are remarkably resistant to damage caused by fast rewarming and reinitiation of the blood flow. Animals in torpor substantially reduce their body temperature and their blood flow; however, they periodically undergo rapid rewarming and restoration of the blood flow during interbouts, which would induce massive damage to tissues under unprotected conditions in non-hibernating species similar to ischemia-reperfusion injury. The protective effect on the tissues of hibernating animals is linked to their altered metabolism largely relying on ketone bodies, such as BHB, and has potential medical implications. A BHB-based therapy was developed for the treatment of hemorrhagic shock in rats [130]. Based on the beneficial impact of BHB and the antioxidant melatonin during torpor and interbout transition, a resuscitation therapy was tested in ischemia-reperfusion animal models of rats and pigs [131,132,133,134]. The use of BHB and melatonin resulted in a significant decrease in mortality following severe hemorrhagic shock, illustrating that the development of hibernation-derived therapies could improve medical treatment strategies for stroke and hemorrhagic shock in the future.

## 7. Mitochondrial Function in Thermogenesis

Thermogenesis in endotherms is an energy-consuming process to maintain the core body temperature in response to cold conditions. Mitochondria are essential for heat production in adaptive thermogenesis. The oxygen consumption rate in mitochondria is proportional to the amount of heat released (as described by Thornton’s rule) because the net chemical reaction for respiration can be considered equivalent to the combustion of organic compounds [135]. According to Thornton’s rule, the heat or energy production of combusted organic compounds is remarkably constant when expressed as energy per mole of oxygen consumed [136]. Fat as an organic compound has a high energy density and produces about twice the heat of carbohydrates when oxidized to carbon dioxide and water. Therefore, lipids serve as an ideal storage and fuel depot in the adipose tissues of animals. Thermogenesis predominantly occurs in brown adipose tissue (BAT) or white adipose tissue (WAT), which undergoes “browning” when exposed to cold. White adipocytes mainly act as a storage depot for triacylglycerols, whereas brown adipocytes or induced “brown like” (beige) adipocytes oxidize lipids in mitochondria to generate heat. Activated beige adipocytes are formed in the WAT during chronic cold conditions. The browning of white adipocytes is promoted by the adipomyokine irisin, which acts in both adipose and muscle tissue, and induces thermogenesis by WAT browning [137,138]. “Classic” or developmentally programmed brown adipocytes originate from a subset of multipotent stem cells, which can differentiate into a skeletal muscle lineage. However, the expression of certain transcriptional regulators, including early B cell factor-2 (EBF2) and PR domain containing 16 (PRDM16), stimulates the differentiation of myoblast progenitor cells toward the brown adipocyte lineage [139,140,141]. Taken together, beige and brown adipocytes are distinct cell types with different origins, although they both play critical roles in cold-induced thermogenesis.

Heat production in brown or beige fat, also known as non-shivering thermogenesis, is mediated by uncoupling protein-1 (Ucp1) in mitochondria. Mitochondria convert the chemical energy of nutrients into electrical voltage. The electrical voltage gradient is primarily established across the inner mitochondrial membrane (IMM) [142]. Ucp1 enables the diffusion of protons across the IMM and thus uncouples the proton gradient from ATP synthesis. Consequently, Ucp1 activity increases the conductance of the IMM to induce heat generation in mitochondria rather than ATP production [1,2]. 

To promote non-shivering thermogenesis in brown or beige fat during chronic cold stress, mitochondria are highly abundant and larger in brown or beige adipocytes and display a packed laminar cristae structure [143]. Mitochondria in thermogenic adipocytes have an elevated oxidative capacity and contain increased levels of enzymes for fatty acid metabolism, the citrate cycle and oxidative phosphorylation [144]. In addition, pyruvate dehydrogenase kinase 4 (PDK4) was found as a highly abundant enzyme in BAT mitochondria. PDK4 is one of the inhibiting kinases for the pyruvate dehydrogenase complex and induces a switch from glucose toward fatty acid oxidation [111].

Adipose tissue is controlled by the sympathetic nervous system. Both BAT and WAT are innervated by sympathetic nerve fibers, which can be tracked back to neuronal circuits that originate from the hypothalamus and the brainstem [145]. Defined hypothalamic sites, including the preoptic area and the dorsomedial hypothalamus, balance different aspects of thermoregulation in BAT for body temperature control or the fever response through downstream innervation circuits [146]. Sympathetic nerve fibers, which innervate adipocytes, release catecholamine, especially norepinephrine, to stimulate β-adrenergic receptor (β-AR) signaling in adipocytes. β-AR signaling activates lipolysis in WAT and induces mitochondrial-based thermogenesis in BAT during cold exposure [147,148,149,150]. A master regulator downstream of β-AR signaling is the transcriptional coactivator PGC-1α, which induces the expression of *Ucp1* and other thermogenic regulators, such as respiratory chain proteins and fatty acid oxidation enzymes [119,151]. PGC-1α is activated by the p38 mitogen-activated protein kinase (p38 Mapk) via phosphorylation (Figure 4) and can be transcriptionally upregulated dependent on the p38 Mapk function and cold exposure [152,153]. Phosphorylated PGC-1α associates as a coactivator with nuclear hormone receptors, such as PPARγ and PPARα, and activates thermogenic genes, including *Ucp1*, in brown adipocytes, as already mentioned above [119,154,155]. In addition, β-AR stimulation in WAT results in elevated lipolysis and increased levels of circulating free fatty acids, which can be taken up by brown and beige adipocytes. In thermogenic adipocytes, free long-chain fatty acids can bind to mitochondrial Ucp1s and increase their proton transport activity (Figure 4) [156]. Taken together, mitochondrial Ucp1 function in thermogenesis is controlled via PGC-1α-mediated transcriptional regulation and by direct stimulation of Ucp1 activity through the binding of free fatty acids.

Cold exposure and the browning of WAT alter the mitochondrial architecture and their interaction with cellular organelles. Cold exposure and adrenergic stimulation of thermogenic adipocytes cause mitochondrial fission [157]. Mitochondrial fission is mediated by dynamin-related protein 1 (Drp1), which is phosphorylated and activated through protein kinase A (PKA) functioning downstream of the β-adrenergic receptor. Drp1 stimulation occurs along with cleavage of the mitochondrial dynamin-like GTPase Opa1 (optic atrophy protein 1), resulting in a simultaneous decrease in mitochondrial fusion upon adrenergic stimulation. Elevated mitochondrial fission and accumulation of mitochondria might improve the accessibility of Ucp1 to certain free fatty acids for Ucp1 activation [157]. Thus, mitochondrial fission likely promotes uncoupling and heat production in brown adipocytes. Intriguingly, mitochondrial dynamics and morphology is controlled by inter-organelle communication with peroxisomes. Peroxisomes are highly abundant in BAT [158] and their number and enzymatic activities increase in a PGC-1α-dependent manner during cold exposure [159,160]. Peroxisomes regulate mitochondrial architecture and their role in thermogenesis via plasmalogens, which are partially synthesized in peroxisomes [161]. Plasmalogens are a sub-class of glycerophospholipids and are present in various cellular membranes, including membranes of mitochondria. Impairment of peroxisomal biogenesis and peroxisomal plasmalogen synthesis results in a fused mitochondrial morphology associated with mitochondrial dysfunction in thermogenesis, whereas dietary supplementation of plasmalogen precursors can restore the mitochondrial architecture and activity in brown and beige adipocytes [161]. Although the mechanism of how plasmalogens regulate mitochondrial functions still needs to be determined, plasmalogens might protect mitochondria through their antioxidative properties [162], and thus, prevent polyunsaturated phospholipids from peroxidation.

## 8. Conclusions and Future Perspectives

Organisms have evolved diverse lipid-metabolism-based strategies to adapt to cold environments. Poikilotherms, which are unable to maintain their internal temperature, rely on sensing changes in membrane rigidity with sophisticated membrane fluidity sensors and can readjust membrane properties to ensure survival during cold exposure. Homeotherms keep their core body temperature constant by inducing thermogenesis in specialized fat tissues to facilitate the oxidation of lipids and heat generation in mitochondria. As an additional adaptation strategy, some species can enter a torpid state to repress their metabolism during periods of limited food supply and cold temperatures. Mechanisms that mediate low-temperature adaptation are linked to very basic animal biology and metabolism; however, they are also highly relevant for a deeper understanding of several human diseases and the design of therapeutic interventions for lipid-related metabolic disorders. Several human disease states are associated with defects in the membrane composition of tissues. Diabetic patients have rigid cellular membranes enriched with saturated fatty acids [163,164,165], likely contributing to the pathophysiology of the disease. Diabetes is also linked to obesity, which is characterized by an imbalance of energy intake and consumption. Thus, targeting BAT to increase energy expenditure by promoting lipid oxidation in obese patients via artificial cold treatment is a promising approach. However, the underlying mechanism of BAT thermogenesis needs to be investigated in more detail because, in addition to cold-induced thermogenesis, an infection-stimulated fever thermogenesis was linked to BAT activity [166,167,168]. It is still under debate whether the febrile response following infection is caused by thermogenesis in BAT, although recent studies suggested that Ucp-1-mediated BAT thermogenesis is dispensable for a lipopolysaccharide (LPS)-induced febrile response in mouse models [169,170]. Hence, for future therapeutic approaches, a more comprehensive understanding of cold versus fever thermogenesis is needed, as well as knowledge of how they could interfere with each other to avoid potential health risks for obese patients treated with artificial cold or thermogenesis-inducing drugs.

In recent studies, the activation of BAT- and Ucp-1-based thermogenesis through cold treatment provided an alternative approach to cancer therapy. Exposure of mice to cold conditions and stimulation of BAT reduced the growth of solid tumors [171]. The tumor-inhibiting effect of BAT activation was further supported by a human pilot study [171]. Individuals exposed to mild cold revealed an elevated BAT function. Reduced glucose uptake into the tumor tissue was observed in a cancer patient following BAT stimulation. This suggests that glucose is predominantly consumed in cold-activated BAT and glucose consumption interferes with the glucose availability for the tumor tissue. For future therapeutic strategies, mild cold exposure of patients or drug-based activation of thermogenesis in BAT could be combined with established anticancer drugs to increase the efficiency of cancer treatments. In conclusion, an improved understanding of mechanisms involved in response to cold stress will be crucial for the development of cold-based therapies and the design of therapeutic interventions for diseases, such as obesity and cancer.

## Figures and Tables

**Figure 2 cells-12-01353-f002:**
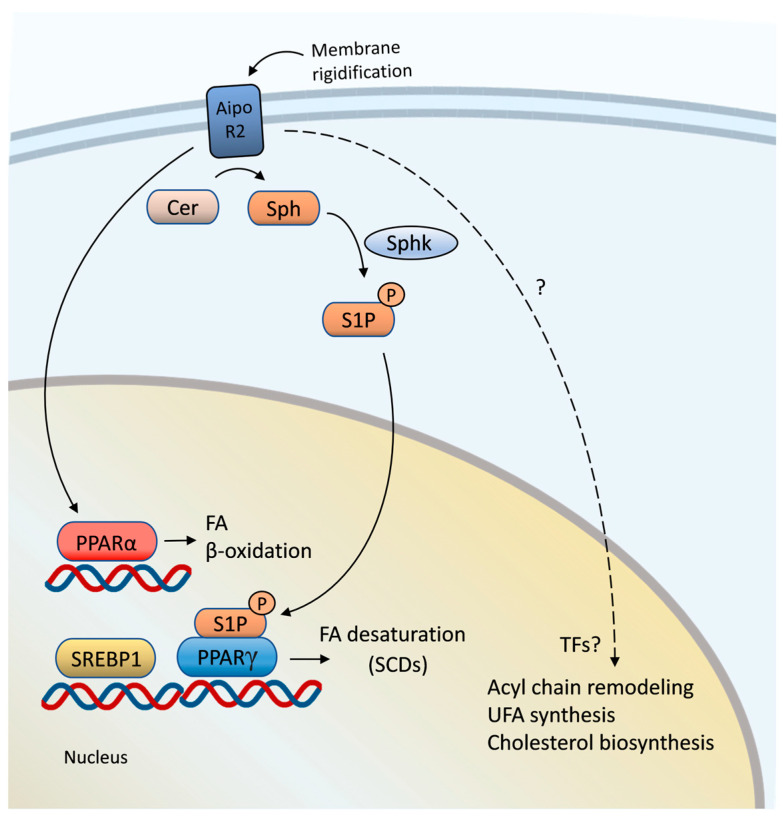
Adiponectin receptor signaling regulates lipid metabolism through PPAR nuclear hormone receptors. The adiponectin receptor AdipoR2 senses membrane rigidification and converts ceramide (Cer) to sphingosine (Sph) via its intrinsic ceramidase activity. Sphingosine is phosphorylated by sphingosine kinases (Sphks) to generate sphingosine 1-phosphate (S1P), which acts as a signaling molecule and induces PPARγ. PPARγ transcriptionally activates stearoyl-CoA desaturases (SCDs) to increase fatty acid desaturation and membrane fluidity. In addition, AdipoR2 can promote PPARα function to enhance fatty acid (FA) catabolism. Upon treatment with saturated fatty acids, AdipoR2 affects the expression of enzymes involved in acyl chain remodeling, UFA synthesis and cholesterol biosynthesis. However, it is not known yet whether these processes are regulated indirectly or by signaling pathways and transcriptional regulators downstream of AdipoR2. P, phosphate; PPARα, peroxisome proliferator-activated receptor-α; PPARγ, peroxisome proliferator-activated receptor-γ; SREBP1, sterol regulatory element-binding protein-1; TF, transcription factor; UFA, unsaturated fatty acid.

**Figure 3 cells-12-01353-f003:**
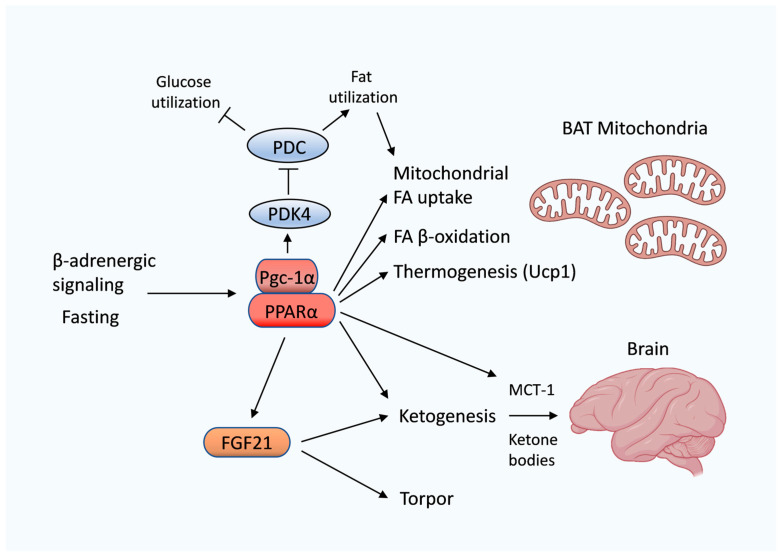
PPARα induces a metabolic shift toward lipid oxidation and ketogenesis in hibernating animals. PPARα promotes lipid catabolism through transcriptionally activating pyruvate dehydrogenase kinase 4 (PDK4), which inhibits the pyruvate dehydrogenase complex (PDC). Suppression of the PDC complex adjusts fuel selection away from glucose and toward fatty acid oxidation. PPARα responds to fasting and β-adrenergic signaling and activates fatty acid (FA) uptake into mitochondria, FA catabolism and Ucp1-mediated thermogenesis. In addition, PPARα induces fibroblast growth factor 21 (FGF21) to promote ketogenesis and torpor. Ketone bodies are elevated during torpor and are an essential fuel source, predominantly for the brain. PPARα directly controls ketone body formation and their transport across the blood–brain barrier through upregulation of the monocarboxylate transporter 1 (MCT-1). BAT, brown adipose tissue; Pgc-1α, PPARγ coactivator-1α; PPARα, peroxisome proliferator-activated receptor-α; Ucp1, uncoupling protein-1.

**Figure 4 cells-12-01353-f004:**
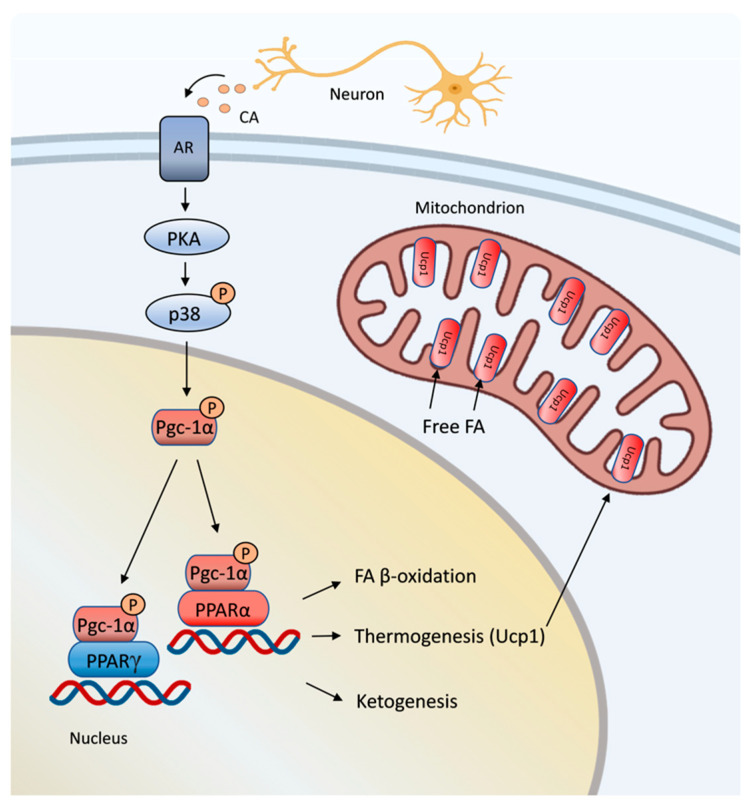
β-adrenergic receptor signaling regulates lipid metabolism and thermogenesis through PPAR nuclear hormone receptors. Following stimulation of the β-adrenergic receptor (AR) with catecholamines (CAs), a PKA–p38-Mapk-based signaling pathway induces the transcriptional coactivator Pgc-1α via phosphorylation. Pgc-1α promotes PPARα and PPARγ function in lipid catabolism, ketogenesis and Ucp1-mediated thermogenesis in mitochondria. FA, fatty acid; P, phosphate; p38 Mapk, p38 mitogen-activated protein kinase; PKA, protein kinase A; Pgc-1α, PPARγ coactivator-1α; PPARα, peroxisome proliferator-activated receptor-α; PPARγ, peroxisome proliferator-activated receptor-γ; Ucp1, uncoupling protein-1.

## Data Availability

Not applicable.

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
