# Peer review of "Molecular Mechanisms of Lipid-Based Metabolic Adaptation Strategies in Response to Cold"

_cells, 2023, doi:10.3390/cells12101353_

Round 1

Reviewer 1 Report

See attached file.

Reviewer 2 Report

Dear editor,

Greeting!

Review of the manuscript: Lipid metabolic adaptation strategies in response to cold by Gang Wu, Ralf Baumeistermembrane and Thomas Heimbucher. This review focuses on how the body regulates its adaptation during cold stress through -bound cold sensors, downstream transcriptional regulators, and lipid metabolism. It is a well-organized and summarized timely review paper because of climate change and artificial factors based on different species.

This review paper promotes our understanding of the mechanism of low temperature adaptation in animals and provides a reference for further research on low temperature adaptation in animals. 

Comments:

1. Cold stress is every common in the natural environment, the effect are also profound and extensive, the review paper focus on the effect including cell / sub-cell structure and metabolism or response. Regretful, it is not to be summarized on the effect of endoplasmic reticulum (ER). Because ER is the locus or organelle where the proteins are translated and folded. So, Please add one sum-tittle to review the strategies on ER under cold stress.

2. “DesK”, Is it an abbreviation? We should give the full name when it first appears.

3. It is strongly agreed that membrane fluidity is important for cell membrane function. However, whether PAQR-2, DesK senses the pressure change inside the membrane, because how the physical property of fluidity is sensed is not explained.

4. Line 181-185: In addition to regulating FA desaturation, AdipoR2 controls cholesterol biosynthesis and acyl chain remodeling of phosphatidylcholines for maintaining membrane homeostasis when cells are challenged with UFAs such as palmitic acid. Membrane rigidifying effects of AdipoR2 deficient cells can be reversed by supplementing membrane-fluidizing MUFAs or PUFAs, suggesting that the primary function of AdipoR2 is to detect and antagonize membrane rigidity.

Here AdipoR2, is a bidirectional catalytic enzyme? So are there other factors that maintain the difference in reactant concentration between the two ends of the reaction?

5. line 264-266: PPARα might promote a shift in fuel utilization from both glucose and fatty acids in regular metabolic conditions towards predominantly fatty acids during hibernation which resembles metabolic changes similar to fasting.

Hibernating animals generally do not feed and are starved, so whether the change in energy use patterns is due to fasting or in response to cold stress.

6. For the figure 2, The implication of the scale and the meaning of the text seem to be contradictory.

7. Calcium, iron and magnesium ions play important roles in lipid metabolism. They also play a role in cold stress? There are non-coding RNAs and so on.

Reviewer 3 Report

In this manuscript entitled “Lipid metabolic adaptation strategies in response to cold”, Wu et al. review organismal responses to detrimental cold stress, and especially focus on lipid metabolism and mitochondrial-related thermogenesis. This review also summarizes the literature on regulatory pathways and their implications in lipid metabolism.

The review is very interesting as cold and heat responses have attracted general interest when we are facing climate change issues. This review integrated some new cellular strategies of molecular mechanisms for the cold response. The writing is good, but also needs to be double-checked carefully to avoid any typos.

I have some suggestions for helping authors to make the manuscript better.

1. In the section “Membrane fluidity and sensing membrane rigidification is essential for cold-adaptation”, would the authors provide a diagram to describe the pathway or processes more clearly?

2. Also in the above section, the content of several paragraphs lacks coherence, and it may be more appropriate to split them into multiple parts.

3. Line 327, why the authors mentioned “hemorrhagic shock” here, does this have anything to do with the above description? Maybe this paragraph is inappropriate in this part.

4. In the section “Mitochondrial function in thermogenesis”, would the authors draw a diagram of thermogenesis related pathways within mitochondria? It is more helpful to understand the relevant situation of thermogenesis in mitochondria.

Reviewer 4 Report

pls see attachement

Reviewer 5 Report

The paper is based on 103 bibliographic items, carefully selected and analysed by Authors and it deals with a topic that is interesting and undoubtedly fills a gap in the works related to organisms cold adaptations.

In my opinion, the work would benefit greatly if Authors expanded it a bit. For example, in the context of the analysis of lower prokaryotic and eucaryotic organisms, Authors focused on mesophilic organisms. What is missing is an analysis of the natural adaptations of microorganisms inhabiting permanently cold environments - psychrophilic/psychrotolerant ones.  In these organisms, the prolonged and continuous pressure of low temperature has caused evolutionary changes in cell structure and metabolism that enable them to survive at low temperatures. Analysing such data and comparing it with those of mesophilic organisms would be of cognitive value and would increase the value of the work.

Round 2

Reviewer 1 Report

There is one sentence in the paper, lines 599-600, that is simply wrong, "Heat generation is based on the Ohmic heating (Joule heating), a process by which the passage of an electric current through a conductor produces heat according to Joule’s first law." Heat is generated by the oxidation reaction which moves electrons from the substrate to O2 and generates H2O and CO2 from the substrate. But this reaction in no way parallels Joule heating by electrons passing through a resistor. Uncoupling from ATP generation makes the oxidation reaction run much faster, but doesn't change the heat per mole of oxygen. The sentence can be deleted without losing anything of note.

The paper can be accepted after this sentence is deleted.

Reviewer 5 Report

I have no comments. Authors have incorporated my suggestions.
In addition, the work benefited after making the changes suggested by the other reviewers.